# Using conventional reinforcement learning algorithms in parameterised action spaces

## Abstract

General purpose reinforcement learning (RL) agents specify exclusively discrete or continuous actions, meaning that tasks with parameterized actions have required bespoke algorithm development. We present a method to convert a parameterised action space Markov decision process into an equivalent Markov decision process with each action being of a simple type. This theoretical insight is developed into a software framework, based on Stable Baselines3 and Gymnasium, which allows researchers to deploy a pair of unmodified standard RL methods where one is responsible for selecting the action and the other for selecting the parameters. Through empirical testing in the Goal and Platform domains we demonstrate algorithm pairings that, with no hyperparameter tuning, achieve comparable performance to the custom-designed and tuned Q-PAMDP and P-DQN. We test this approach in two domains - Robot Soccer and Platform - and compare against the bespoke existing approaches for these two domains.

## 1 Introduction

The process of reinforcement learning (RL) (Sutton & Barto, 2018) produces an approximation to the optimal policy for a given Markov Decision Process (MDP). However real world scenarios often feature exceptionally large action spaces such that the "curse of dimensionality" (Bellman, 1966) limits the rate of convergence to an optimal policy. To offset this complication, structured actions can be specified when defining the model itself.

One example of this approach is the Parameterised Action-space MDP (PAMDP) (Masson et al., 2016). In this setting each discrete *action* chosen requires an associated continuous *action-parameter* selection, determining "what to do" and "how to do it" respectively. These are submitted simultaneously by the agent to the environment, as a pair, to form a single action at each timestep. The most popular PAMDP's in the literature are Platform (Masson et al., 2016), in which the agent must learn to perform in a computer game with actions such as "run" and "leap" and parameters such as direction and speed, and Goal (Kitano et al., 1997), a simulated soccer environment with actions including "shoot" and "run", and parameters again including direction and speed. Both environments have been solved to an extent by bespoke algorithms designed and implemented specifically for the environment, restricting the generalisability of these methods.

The reason that bespoke algorithms have been developed and deployed is that existing general purpose methods for learning purely discrete or continuous action policies are unsuitable for learning parameterised action policies. Approaches to customise either methods or environments typically include mapping the action space to a purely continuous space (Hausknecht & Stone, 2016), or alternating between updating a discrete policy and continuous policy in a bespoke and highly coupled manner (Masson et al., 2016).

In this article we introduce a general purpose decomposition of PAMDPs into an equivalent MDP where at each time point we select either an action or an action-parameter, but not both. We go on to demonstrate this in practice, implementing a Gymnasium (Towers et al., 2023) wrapper to provide an MDP-based interface to any PAMDP-based environment. Forgoing the need for case-by-case engineering, we then introduce a modular class of hybrid agents, composed of arbitrary pairs of pre-existing "Stable Baselines 3 (SB3)"-class (Raffin et al., 2021) RL agents (one discrete, and one continuous per pair). To benchmark the performance of these hybrid agents, we trial various

compositions across two widely known PAMDP-based environments (namely the aforementioned Platform and Goal).

By enabling the reuse of existing reinforcement learning algorithms, our method ensures that developments in conventional reinforcement learning can immediately be reused in the parameterised action space case, thereby raising the ability to find solutions to PAMDP's. Additionally, the modular nature of our implementation enables rapid experimentation to identify optimal hybrid policy compositions for parameterised action-space reinforcement learning.

## 1.1 RELATED WORK

Masson et al. (2016) originally introduced the concept of a PAMDP, and the Q-PAMDP algorithm to solve it, utilising "direct alternating optimisation" which alternates between updating both a discrete-action policy and a parameter policy in turn. This method is not modular, relying on a highly coupled bespoke implementation. A simpler direct policy search method is introduced and demonstrated using "eNAC". Tests are conducted on two domains — Robot Soccer (Kitano et al., 1997) and Platform (the latter itself the author's contribution) — to demonstrate the algorithms' performances, with Q-PAMDP proving superior. Additionally the authors prove that Q-PAMDP converges to a local optimum.

Hausknecht et al. (2016) introduces a freely available implementation of the Half Field Offense (HFO) domain which features a parameterised action space akin to that described by Masson et al. (2016). Building on this, Hausknecht & Stone (2016) introduce the PADDPG algorithm - an actor-critic method based on DDPG (Lillicrap et al., 2016) which works by relaxing the parameterised action space into a continuous approximation for the critic, whilst producing an actor which simultaneously chooses both a discrete action and associated parameter. This simultaneous decision making is highlighted as a key difference of PADDPG from Masson et al.'s Q-PAMDP. Notably, the authors do not provide an empirical comparison of Q-PAMDP's performance to compare against that of PADDPG, potentially due to the long training time spent to train each agent in their results.

Also building on Masson et al.'s work, Wei et al. (2018) introduce PATRPO and PASVG(0) (based on TRPO Schulman et al. (2015) and SVG (Heess et al., 2015) respectively), comparing these algorithms with PADDPG in both the HFO and Platform domains. The results demonstrate unstable yet improved return when using PATRPO over PADDPG or PASVG(0).

Xiong et al. (2018) introduce two new domains: "Simulation" which involves the movement of a point mass in two dimensions towards a target (with two available variants referred to as "Moving" and "Sliding" respectively (Hirtz, 2022)), and "King of Glory" which is a MOBA[1] game with 200 million monthly active players as of July 2017. They also introduce the P-DQN algorithm, alongside an asynchronous variant, and test it in the two new domains in addition to the standard HFO domain. The P-DQN algorithm's performance is compared against PADDPG (Hausknecht & Stone, 2016) and DQN (Mnih et al., 2015) (with the latter only tested in HFO using a simplified discrete action space instead). The authors note that P-DQN converges faster and to a more stable policy than the two other methods tested.

## 2 MATHEMATICAL FRAMEWORK

A PAMDP consists of a set of **states** $s \in S$, and a set of (composite) **actions**

$$A = \bigcup_{a_d \in A_d} \{(a_d, \theta) \mid \theta \in \Theta_{a_d}\},$$

where $A_d$ is a discrete set of "actions" and for each $a_d \in A_d$ the set $\Theta_{a_d} \subseteq \mathbb{R}^{m_{a_d}}$ is the set of "action-parameters". In addition we have the standard MDP components:

**Reward function** $r : S \times A \to \mathbb{R}$

**Transition function** $p : S \times A \times S \to [0, 1]$

**Discount factor** $\gamma \in [0, 1]$

---

[1]https://en.wikipedia.org/wiki/Multiplayer_online_battle_arena

To act in a PAMDP, we consider a policy

$$\pi(a_d, \theta \mid s) = \pi_\omega^d(a_d \mid s)\pi_\psi(\theta \mid s, a_d),$$

the product of a "discrete-action" policy $\pi_\omega^d$, and an "action-parameter" policy $\pi_\psi$, where $\omega$ and $\psi$ denote parameters of the respective policies.

The expected future discounted reward starting in state $s$ and following the policy determined by $\omega, \psi$ is given by $V_{\omega,\psi}(s)$ solving Bellman's equation:

$$V_{\omega,\psi}(s) = \mathbb{E}_{a_d \sim \pi_\omega, \theta \sim \pi_\psi}\Big[r(s, (a_d, \theta)) + \gamma \sum_{s' \in S} p(s, (a_d, \theta), s')V_{\omega,\psi}(s)\Big].$$

Optimal policy parameters $(\omega^*, \psi^*)$ satisfy $V_{\omega^*,\psi^*}(s) = \max_{\omega,\psi} V_{\omega,\psi}(s) =: V^*(s)$; to ease exposition of this article we assume a solution to this optimisation exists, rather than contend with finding the parameters which minimise the Bellman gap, as is necessary if no solution exists.

## 2.1 DECOMPOSED MDP

We decompose this framework into an MDP which represents alternating between action selection and action-parameter selection. We will show that the value function of the decomposed MDP is the same as the value function of the PAMDP. But we will then have an MDP, which will allow us to deploy conventional MDP-based algorithms against PAMDPs, allowing for both faster development of solutions in PAMDPs, and comparison of PAMDP-specific algorithms with standard RL algorithms.

Our approach is to double the number of time steps, with each step of the PAMDP corresponding to two steps of an MDP. We select the action on odd timesteps, and the corresponding action-parameter on even timesteps. This method of conversion can be paired with a hybrid policy, formed by pairing any two conventional discrete and continuous reinforcement learning algorithms respectively, and exposing each to one of two component MDP "views" whose dynamics are based in part on the other policy, and whose action space is either entirely discrete or continuous in nature.

We now show that the resulting MDP has an identical value function as the original PAMDP, meaning that optimal strategies of the MDP transfer directly to being optimal strategies of the PAMDP.

Consider a PAMDP $(S, A, r, p, \gamma)$. We form the decomposed MDP $(\tilde{S}, \tilde{A}, r, p, \tilde{\gamma})$ by specifying each component in the following manner:

- $\tilde{S} := S \cup S_\theta$ where $S_\theta := S \times A_d$, so that as well as the original states in $S$ we also have states $(s, a_d)$ which enhance the original states with the action selection.

- $\tilde{A} := A_d \cup A_\Theta$ where $A_\Theta := \bigcup_{a_d \in A_d} \Theta_{a_d}$ are the actions from states $s_\theta = (s, a_d) \in S_\theta$. Clearly not all actions are valid in all states, and we simply mask off the invalid actions in order to avoid notational overload.

- Given $\tilde{s} \in \tilde{S}$ and $\tilde{a} \in \tilde{A}$, define

$$\tilde{r}(\tilde{s}, \tilde{a}) := \begin{cases} 0 & \text{if } \tilde{s} \in S \\ \gamma^{-\frac{1}{2}} r(s, (a, \theta_a)) & \text{if } \tilde{s} = (s, a) \in S_\theta, \tilde{a} = \theta_a \in \Theta_a \end{cases}$$

- Given $\tilde{s}, \tilde{s}' \in \tilde{S}$, and $\tilde{a} \in \tilde{A}$, define

$$\tilde{p}(\tilde{s}, \tilde{a}, \tilde{s}') := \begin{cases} 1 & \text{if } \tilde{s} \in S, \ \tilde{a} \in A_d, \ \tilde{s}' = (\tilde{s}, \tilde{a}) \\ p(s, (a, \theta_a), \tilde{s}') & \text{if } \tilde{s} = (s, a) \in S_\theta, \ \tilde{a} = \theta_a \in \Theta_a, \ \tilde{s}' \in S \\ 0 & \text{otherwise} \end{cases}$$

- $\tilde{\gamma} = \gamma^{\frac{1}{2}}$

When a discrete action is selected, we allocate no reward, and transition to the "enhanced" state consisting of the origin state and the discrete action. When an action-parameter is selected in enhanced state $(s, a_d)$ we allocate a corrected reward, and transition to the state that the PAMDP would transition to from state $s$ when supplied with the the discrete action component of the enhanced state, $a_d$, and the selected action-parameter. The discount factor $\tilde{\gamma}$ in the decomposed MDP is the square root of the original discount factor $\gamma$ since it is applied twice as often in the MDP as in the PAMDP.

**Proposition 1.** *The decomposed MDP $(\tilde{S}, \tilde{A}, \tilde{r}, \tilde{p}, \tilde{\gamma})$ has policy value function $\tilde{V}_{\omega,\psi}$, and optimal value function $\tilde{V}^*$, that satisfy $\tilde{V}_{\omega,\psi}(s) = V_{\omega,\psi}(s)$ and $\tilde{V}^*(s) = V^*(s)$ for all $s \in S$.*

*Proof.* Start by considering the $\tilde{V}_{\omega,\psi}$ of the decomposed MDP for fixed policy $\pi_{\omega,\psi}$. Consider an arbitrary state $s \in S$. We have that

$$\tilde{V}_{\omega,\psi}(s) = \mathbb{E}_{a_d \sim \pi_\omega} \left[ \tilde{r}(s, a_d) + \tilde{\gamma} \sum_{\tilde{s}' \in \tilde{S}} \tilde{p}(s, a_d, \tilde{s}') \tilde{V}_{\omega,\psi}(\tilde{s}') \right]$$

$$= \mathbb{E}_{a_d \sim \pi_\omega} \left[ 0 + \gamma^{\frac{1}{2}} \tilde{V}_{\omega,\psi}((s, a_d)) \right],$$

since no reward is given from states $s \in S$ and transition to $\tilde{s}' = (s, a_d)$ is deterministic given $a_d$.

Now consider the term for the enhanced state:

$$\tilde{V}_{\omega,\psi}((s, a_d)) = \mathbb{E}_{\theta \sim \pi_\psi} \left[ \tilde{r}((s, a_d), \theta) + \tilde{\gamma} \sum_{\tilde{s}' \in \tilde{S}} \tilde{p}((s, a_d), \theta, \tilde{s}') \tilde{V}_{\omega,\psi}(\tilde{s}') \right]$$

$$= \mathbb{E}_{\theta \sim \pi_\psi} \left[ \gamma^{-\frac{1}{2}} r(s, (a_d, \theta)) + \gamma^{\frac{1}{2}} \sum_{s' \in S} p(s, (a_d, \theta), s') \tilde{V}_{\omega,\psi}(s') \right]$$

where again the summation range is restricted to the states for which the transition probability is non-zero, which this time is the original non-enhanced states of the PAMDP.

Combining these two expressions, we see that for $s \in S$

$$\tilde{V}_{\omega,\psi}(s) = \mathbb{E}_{a_d \sim \pi_\omega, \theta \sim \pi_\psi} \left[ r(s, (a_d, \theta)) + \gamma \sum_{s' \in S} p(s, (a_d, \theta), s') \tilde{V}_{\omega,\psi}(s') \right].$$

Since this is the Bellman equation for the original PAMDP, we see that $\tilde{V}(s) = V(s)$ for all $s \in S$.

Since $\tilde{V}^*(s)$ and $V^*(s)$ are both obtained from $\tilde{V}_{\omega,\psi}(s)$ and $V_{\omega,\psi}(s)$ by maximising over $\omega$ and $\psi$, we see that also $\tilde{V}^*(s) = V^*(s)$ for all $s \in S$. $\qquad\square$

We have demonstrated how to construct an associated MDP for any given PAMDP, with a shared optimal value function. An optimal policy for one process is also optimal for the other. Hence learning an optimal policy for the decomposed MDP is sufficient to solve the PAMDP. We therefore proceed to develop methods to automatically convert PAMDPs into the decomposed MDP, and deploy standard reinforcement learning methods to solve the MDP.

## 3 SOFTWARE IMPLEMENTATION

We introduce a modular and extensible framework for conducting reinforcement learning experiments in parameterised action spaces, with a strong emphasis on clarity, reproducibility, and flexibility. Its defining contribution lies in a "converter" wrapper and an associated "hybrid policy" object. The hybrid policy is a novel method of combining two distinct reinforcement learning agents: one operating over discrete actions, the other over continuous action-parameters. This hybrid policy is then trained using an MDP-based environment derived (using the converter) from a given PAMDP-based environment for compatibility with otherwise incompatible discrete and continuous action-space algorithms.

This sophisticated architecture is designed specifically to ensure existing reinforcement learning algorithms may be applied to the parameterised action space setting with minimal overhead, supporting the reuse of new state of the art algorithms which would otherwise require extensive bespoke engineering to achieve. In this way we enable fast experiment design and iteration to explore novel approaches within this problem setting. The system, described in Fig. 1, takes as input a given Gymnasium-based environment whose parameterised action space is modelled as a gymnasium `Tuple` containing two objects - a `Discrete` space and a second Gymnasium tuple of `Box` spaces - representing the discrete action space and continuous action-parameter spaces respectively. PAMDP-based Gymnasium environments not modelled in this manner may utilize an observation wrapper to achieve compatibility with our system.

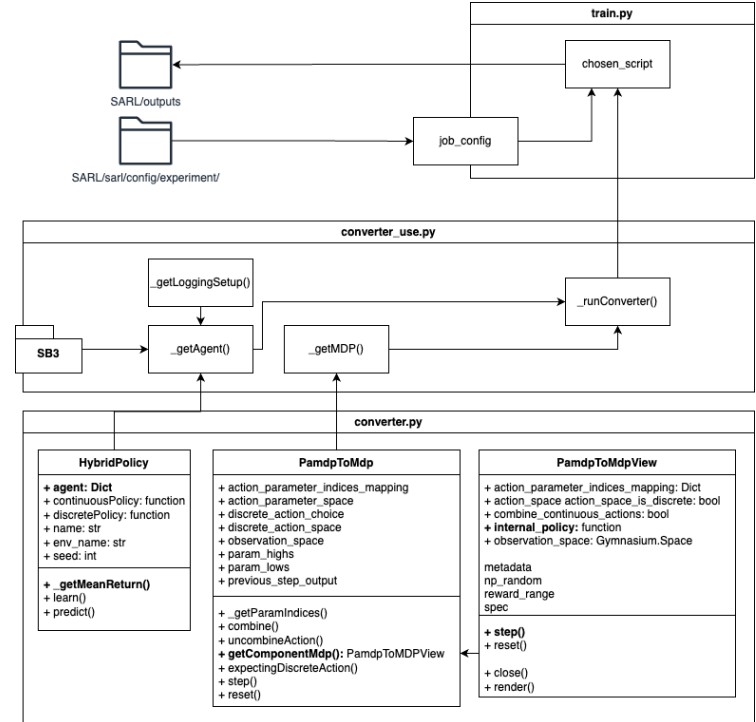

Figure 1: Class diagram for the framework's converter usage. The train.py file is provided a specific function from converter_use.py based on the user-specified job_config. The converter.py file contains the three components discussed in detail.

This method of agent composition is made possible through three novel components which construct a single policy from the prediction output of the two component agents, and which derive an MDP interface to the PAMDP by intelligently deciding which action space is presented to the composite agent. The result is a training process which utilises the `.learn` method of each Stable Baselines3 agents in an alternating manner taking actions within a derived gymnasium environment which acts as an interface to the converted PAMDP. To explain this we explore the components individually in turn below.

### 3.1 COMPONENTS

Our hybrid reinforcement learning approach is enabled by a coordinated interaction between three key software components: the PAMDP-to-MDP converter (`PamdpToMdp`), the `HybridPolicy` class, and the `PamdpToMdpView` abstraction, illustrated in Fig. 1.

### 3.1.1 PAMDP-TO-MDP WRAPPER (`PamdpToMdp`)

Our core innovation is a Gymnasium-compatible wrapper that transforms a PAMDP into a form accessible by conventional MDP-based agents. To manage the alternating nature of discrete and continuous decisions, the wrapper augments each state observation with an additional variable $a_d$. This variable encodes the expected action type:

1. If $a_d = -1$, a discrete action is expected.
2. If $a_d \geq 0$, the agent is expected to provide the continuous parameter for discrete action $a_d$.

Upon receiving a discrete action $a_d$, the wrapper caches it internally, returns zero reward, and returns the augmented state, ready to receive the corresponding action-parameter in the following timestep. When an action-parameter is received, it is submitted together with the discrete action as a single perceived action to the wrapped PAMDP. Output from the PAMDP is then returned to the

accessing agent, ensuring a seamless interaction despite the difference in perceived timesteps and expected inputs/outputs.

### 3.1.2 COMPONENT MDP VIEWS (`PamdpToMdpView`)

The purpose of our work is to allow the use of standard SB3 agents to solve parameterised action space reinforcement learning problems. When we instantiate the combined agent to learn the problem, we therefore need to instantiate two component SB3 agents. Namely, one agent to select from the discrete actions, and one agent to select the associated action-parameter.

Unfortunately SB3 agents require fully defined MDPs with consistent action spaces at the point of instantiation. To meet this requirement, the `PamdpToMdp` wrapper offers partial MDP views via the `PamdpToMdpView` class. These act as partial representations of the full PAMDP, exposing either the discrete or continuous action-space independently. Internally, they interface with the shared environment state but present SB3-compatible surfaces, allowing conventional agents to operate seamlessly together on otherwise incompatible spaces. This is achieved by passing the action selection method of each agent to the other agent's `PamdpToMdpView`. The view can then use the current policy of the other agent to form each parameterised action when receiving action submissions from the learning agent. In other words, first the discrete action learner can update its policy whilst acting within its environment view (where the view obtains its action-parameter selections from the action-parameter learner. This is followed by the action parameter learner updating its policy via its own environment view, whilst the now temporarily fixed discrete action policy provides the missing components for each parameterised action submission.

### 3.1.3 HYBRID POLICY COMPOSITION (`HybridPolicy`)

To enable training over the full PAMDP, we define a `HybridPolicy` class that composes a discrete and a continuous agent. When instantiated, this class holds references to two preconfigured SB3 agents and provides a unified `.learn()` interface, though the policy object is not itself an SB3 agent. Training is conducted internally in cycles using the aforementioned `PamdpToMdpView`'s, where the total number of timesteps is divided among discrete and continuous learners according to a user-defined parameter. During each cycle, the wrapper alternates between training the discrete agent on the discrete MDP view and the continuous agent on the continuous MDP view, thereby enabling both components to gradually converge and forge the learned hybrid policy.

In selecting action-parameters, one continuous component agent is used across all action-parameter spaces, which are presented as a single composite continuous space to this agent. The agents' outputs are masked to obtain only the relevant action-parameter information at each stage.

## 4 EXPERIMENTS

We carry out experiments in two standard PAMDP environments, and compare two baseline custom-built PAMDP algorithms with a set of approaches enabled by the introduction of our framework. We explore the performance of various SB3-learned policies enabled by our framework, comparing against the two baselines Q-PAMDP (Masson et al., 2016) and P-DQN (Xiong et al., 2018).

We evaluate agents across Gymnasium implementations of the two aforementioned standard benchmarking environments `Platform-V0` (Masson et al., 2016) and `Goal-V0` (a derivative of the robot soccer domain; Kitano et al., 1997)[2]. Both environments are chosen as they subclass `Gymnasium.Env`, ensuring compatibility with common reinforcement learning interfaces, each notably featuring a shared grammar in their method of representing a parameterised action-space (no consensus standard representation exists at time of writing).

To guarantee reproducibility across different computing setups, we employ `Poetry` for dependency resolution and virtual environment management. This ensures that all required packages, along with their versions, are consistently installed across platforms. Coupled with `Hydra`, this allows users to define experiments either through inline CLI arguments or prewritten YAML files. Each experiment

---

[2]These environments and all other code will be released in the project's Github repository (link to be added later).

Table 1: Performance on `Platform-V0` (mean best learned policy ± standard deviation, 50 runs)

| Continuous: | | A2C | DDPG | PPO | SAC | TD3 |
|---|---|---|---|---|---|---|
| | A2C | 0.21 ± 0.11 | 0.22 ± 0.09 | 0.14 ± 0.11 | 0.40 ± 0.27 | 0.19 ± 0.07 |
| Discrete: | DQN | 0.30 ± 0.06 | 0.79 ± 0.22 | 0.08 ± 0.02 | 0.91 ± 0.02 | 0.79 ± 0.22 |
| | PPO | 0.11 ± 0.01 | 0.30 ± 0.31 | 0.34 ± 0.07 | **0.98 ± 0.06** | 0.24 ± 0.02 |
| Baselines: | | QPAMDP: | 0.76 ± 0.02 | PDQN: | 0.97 ± 0.00 | |

configuration is automatically saved alongside its corresponding results in the `.hydra` directory. This preserves the complete experiment specification, facilitating exact reruns at any point in the future.

We execute the experiments in parallel on Lancaster University's High-End Computing Cluster (HEC), with each job assigned a single NVIDIA V100 GPU and 4GB of RAM. Each (policy, environment) pair, combined with an integer seed, constitutes one unique job submission, supporting efficient multi-trial experimentation across seeds and hybrid policy compositions. Each (policy, environment) combination is run $n = 50$ times to allow averaging across trials.

The combinations to be tested are composed from three discrete-capable algorithms (PPO, A2C, and DQN (Schulman et al., 2017; Mnih et al., 2016; 2015)), five continuous-capable algorithms (PPO, A2C, DDPG, SAC, and TD3 (Schulman et al., 2017; Mnih et al., 2016; Lillicrap et al., 2016; Haarnoja et al., 2018; Fujimoto et al., 2018)), and the Platform and Goal environments.

Performance of each algorithm is measured during learning by pausing the learning process after each cycle of updating continuous parameters and discrete parameters, and measuring the performance of the current learned policy in the base environment. The reason for this is to avoid complications arising from using two SB3 algorithms in parallel. For our final performance of the learning run, we take the highest scoring policy that arose across all trials during learning.

### 4.1 PLATFORM

The platform domain is a novel domain introduced in Masson et al. (2016), modelling the action-selection and reward experience of a player playing a video game of the platforming genre.The state is composed of 4 variables representing agent position $x$, agent speed $\dot{x}$, enemy position $ex$, and enemy speed $e\dot{x}$. The agent experiences a constant negative vertical acceleration when not on the ground, during which time their actions have no effect; the singular enemy moves in an unspecified manner when the agent is on their platform, and is otherwise stationary; the episode ends when the agent reaches the goal platform, makes contact with the enemy, or falls below the height of the platforms. Two primitive actions — run and jump — become a 3-action parameterized action space $A = \{\mathtt{run}(\theta_1), \mathtt{hop}(\theta_2), \mathtt{leap}(\theta_3)\}$, where `hop` and `leap` represent two different kinds of jumps, and each parameter represents an associated magnitude. The reward at time-step $t$ is specified as $r_t = \frac{|x_t - x_{t+1}|}{l}$ where $l$ is the length of the current episode's level.

The results from the Platform-V0 environment provide a compelling validation of the modular hybrid policy framework. As detailed in Table 1, which summarizes the performance metrics from the experiment, several of the hybrid policy compositions demonstrate comparable performance to the best of the established baselines. Namely these are DQN-SAC, PPO-SAC, (and then to a lesser extent) DQN-TD3, and DQN-DDPG; we plot these algorithms' performance with the baselines' in Fig. 2. Note that even before undergoing the hyperparameter optimisations found in the baseline cases, the general-purpose SB3 RL algorithms perform comparably to algorithms specifically engineered for this problem type.

### 4.2 ROBOT SOCCER

The first of the two domains used in Masson et al. (2016), Robot Soccer (itself a simplified version of the domain found in RoboCup; Kitano et al., 1997) models an agent playing a game of soccer attempting to score a goal against an adversarial goalkeeper.

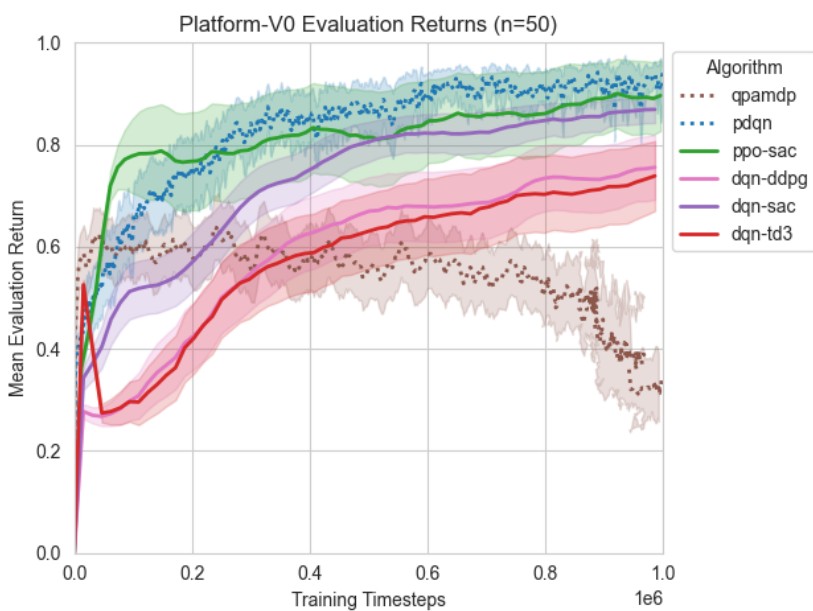

Figure 2: Time-windowed moving average of the average performance of the top performing composite policies and the two baseline algorithms for the `Platform-V0` environment.

Table 2: Performance on `Goal-V0` (mean best learned policy ± standard deviation, 50 runs)

| Continuous: | | A2C | DDPG | PPO | SAC | TD3 |
|---|---|---|---|---|---|---|
| | A2C | -6.49 ± 8.44 | 1.35 ± 9.63 | -2.66 ± 11.1 | 6.87 ± 6.16 | -0.15 ± 7.79 |
| Discrete: | DQN | -8.82 ± 1.79 | 4.28 ± 11.43 | -17.2 ± 0.33 | **8.79 ± 11.11** | 3.82 ± 10.86 |
| | PPO | -2.97 ± 5.84 | -3.06 ± 6.34 | -2.71 ± 4.09 | 2.42 ± 8.52 | -6.35 ± 1.41 |
| Baselines: | | QPAMDP: | -6.72 ± 0.25 | PDQN: | 22.75 ± 5.22 | |

The state available to our agent $S$ is composed of the agent, goalkeeper, and ball's respective positions and velocities, in addition to the orientations of the agent and goalkeeper. The result is 14 continuous state variables such that $S = \{(x_n, y_n), (r_n, \phi_n), \alpha_m \mid n \in \{1, 2, 3\}, m \in \{1, 2\}\}$. When not in possession of the ball, the agent automatically moves towards the ball until it has possession. The keeper moves towards a stationary ball, or towards the ball's future location when it is in motion. If the keeper takes possession of the ball, or the ball either reaches the goal or leaves the field, the episode ends.

The action space consists of two action types - kick-to$(x, y)$ and shoot-goal$(h)$ - representing kicking the ball to a given position $(x, y)$, and kicking the ball towards some position $h$ along the goal line respectively. The authors split the latter action into two, shoot-goal-left$(h_L)$ and shoot-goal-right$(h_R)$, to induce a bias towards faster learning of the optimal policy (as the author's note such a policy is discontinuous, as at no point should the agent shoot the ball towards the keeper). The result is a 3-item parameterized action space of $A = \{\text{kick-to}(x, y), \text{shoot-goal-left}(h_L), \text{shoot-goal-right}(h_R)\}$.

A reward of 0 is provided for all non-terminal actions. Terminating with a goal gives a reward of 50. Failing to score is given a negative reward, $R = -d$, where $d$ is the distance from the goal.

From Table 2 we observe a wide disparity between the PDQN baseline and all other algorithms, including the QPAMDP baseline also. This is only further emphasised through Fig. 3. In this more complex setting it becomes clear the value of fine tuning, whereby the tailored hyperparameter

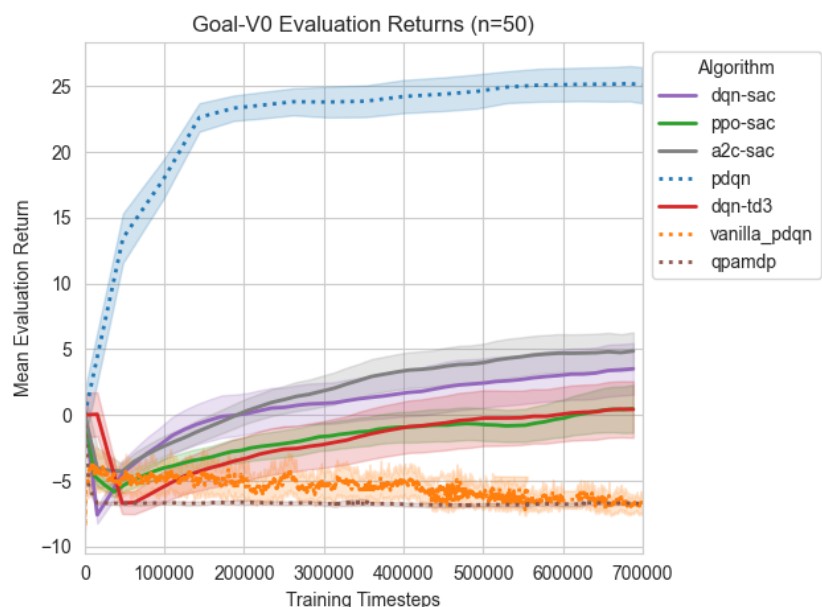

Figure 3: Time-windowed moving average of the average performance of the top performing composite policies and the two baseline algorithms for the `Goal-V0` environment.

choices and initial values inherent to PDQN's experiment configuration significantly aid in the rate of its convergence towards an optimal policy, and the magnitude of returns its successive policies can obtain. Notably, further exploratory PDQN trials without user-provided domain knowledge in the form of initial action weights and biases resulted in dramatically reduced evaluation returns (less than 0) therein demonstrating this baseline's limitations compared to our work.

We can observe in Figure 3 that all of the best performing algorithm combinations are continuing to rise throughout the training timesteps. This would suggest that each of DQN-SAC, PPO-SAC, A2C-SAC and DQN-TD3 may attain a greater maximum evaluation return if assessed over a longer training period. We also observe the poor performance of 'vanilla PDQN', where it has not been supplied with custom initial values. Lastly we can observe Q-PAMDP's inability to perform desirably in this setting, even with domain-knowledge based initialisation weights.

## 5    CONCLUSIONS

We have presented a method for solving parameterised action-space MDPs using appropriate combinations of standard reinforcement learning algorithms for either discrete or continuous action-spaces. The decomposition to a standard MDP with alternate discrete and continuous action selections is shown to have an identical optimal policy to the original PAMDP. The implementation of this approach is detailed, outlining how a pair of Stable Baselines3 agents may form a hybrid policy, each provided one of two MDP "views" within which to learn. Each view acts as an interface to the converted PAMDP, necessary to satisfy the requirements of SB3 agents themselves.

Our experiments demonstrate that pairs of Stable Baseline agents can outperform the state of the art with no hyperparameter tuning in the `Platform-V0` environment, and perform moderately well with no hyperparameter tuning in the `Goal-V0` environment. However they are outperformed by a highly tuned version of the PDQN baseline on this more complex environment.

Our general purpose framework, with the classes `PamdpToMdp`, `PamdpToMdpView` and `HybridPolicy` all acting generically on `Gymnasium`-compliant PAMDP environments, presents an opportunity for more rapid development of solutions to PAMDP problems.

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
