# OpenReview forum: "Using conventional reinforcement learning algorithms in parameterised action spaces"
_ICLR.cc/2026/Conference — Submitted to ICLR 2026_

### Official Review · Reviewer_bosw · 2025-10-29

**Soundness:** 3
**Presentation:** 2
**Contribution:** 2
**Rating:** 2
**Confidence:** 4

**Summary:**

This paper proposes to solve parametrized-action MDPs (PAMDPs) by converting them to an MDP format where an agent must take two actions in a row, without reward or any dynamics: first a discrete, then a continuous, action. The second action depends on an expanded state space which includes the selected discrete action. The authors prove that one can retrieve the optimal optimal value function in the new MDP can be trivially converted to an optimal policy in the PAMDP. This has the advantage of allowing us to use off-the-shelf non-PAMDP algorithms for the two components, rather than having to design specialized PAMDP algorithms. The authors demonstrate this using various combinations of off-the-shelf Stable Baselines 3 agents in two standard PAMDP domains.

**Strengths:**

- This idea is novel, and interesting - I like the perspective, and I think it opens up a wide range of very practical PAMDP solutions, which is what we need.
- The proof of equivalent is simple but makes the point clear.
- The writing is mostly clear and the idea is well-communicated.
- If all the other weaknesses were resolved (see below), I think this would be a strong paper - because of, not despite, the simplicity of the idea, which is admirable.

**Weaknesses:**

Unfortunately, I think the paper displays several weaknesses:

1. Badly calibrated ideas about generality
 - The authors use "general purpose RL" to mean RL problems with one action type or the other, and refer to PAMDPs as a more specific special case. Actually PAMDPs are more general than either case. There happen to be a lot of algorithms available for the single-action-type case, but mathematically either discrete or continuous actions are less general than the PAMDP case.
 - The authors misuse the word "bespoke" to imply that e.g. the QPAMDP algorithm was designed especially for *a domain*. But what they actually mean is that the algorithm was designed for the entire very general PAMDP setting. That is the opposite of a bespoke algorithm. An algorithm which can be reasonably applied to any PAMDP is not in any way bespoke. The claim that they require "case by case engineering" is puzzling. Perhaps the authors mean hyperparameter tuning? Also, it's not clear that the "highly coupled manner" in which PAMDP action selection works is a bad thing at all. Why should it be? The actions and their parameters *are* highly coupled.
 - Any actual instance of the authors's framework requires selecting specific algorithms for the discrete and continuous part. Though the framework is general, it is also not an algorithm - any such specific algorithm must require these choices, and so would therefore be what the authors consider "bespoke". In fact, the QPAMDP algorithm that the authors compare to is (barring some minor adjustments) an instance of their framework, with Sarsa-lambda used for the discrete action selection and natural actor critic for continuous action selection. These two choices just happen to be dated (and not, in the original paper, Deep).

2. Outdated baselines.
 - The authors compare to QPAMDP, which is nearly a decade old at this point, and PDQN, which is a reasonable benchmark. But the most recent relevant algorithm is DLPA. published at ICML last year by Zhang et al. That paper also includes experiments showing that HyAr is probably a better-performing algorithm than PDQN. So at least comparisons to HyAR and DLPA are mandatory. That paper includes public source code, so all of its baselines should be easy to replicate.

3. Limited experimental support
 - The authors use only two baselines, whereas the recent DLPA paper includes eight.
 - Their experiments, even varying across several Stable Baseline methods, are either not much stronger than PDQN, or demonstrably worse.

4. Implementation included in the main body of the paper
 - The authors spend a substantial portion of the paper talking about their code. There is even a class diagram. I cannot emphasize strongly enough how the scientific merit of a paper has absolutely nothing to do with its code, or the software frameworks it was built in or exploits, all of which will be obsolete and forgotten long before (we hope) this paper is still being read. The science, ideas, and math in the paper must stand alone and be explained without reference to these implementation details, which often obscure those ideas.

**Questions:**

I would like to see comparisons to DLPA and HyAR, one at least the domains in the DLPA paper.

---

### Official Review · Reviewer_EZWs · 2025-10-29

**Soundness:** 2
**Presentation:** 2
**Contribution:** 2
**Rating:** 4
**Confidence:** 4

**Summary:**

This paper proposes a method for transforming parameterised action-space MDPs (PAMDPs) into equivalent standard MDPs, enabling the use of existing discrete and continuous reinforcement learning (RL) algorithms to handle hybrid-action problems. The approach is formalized through a proof of policy equivalence between the original PAMDP and its decomposed version and implemented as a modular framework integrated with Gymnasium and Stable Baselines3 (SB3). The authors benchmark combinations of standard algorithms (DQN, PPO, SAC, etc.) against dedicated PAMDP baselines (Q-PAMDP and P-DQN) on the Platform and Goal environments.

**Strengths:**

The paper provides a mathematically sound proof that the proposed decomposition preserves optimal policies, which supports the theoretical validity of the approach.

The framework design is well documented, allowing hybrid-action experiments with standard RL agents without custom implementations.

Empirical results indicate that simple SB3 algorithm combinations can perform on par with specialized PAMDP methods in some cases, highlighting the framework’s practical usability.

The manuscript is clearly structured and easy to follow, with well-defined notation and a consistent exposition of concepts.

**Weaknesses:**

Lack of analysis of the alternating training procedure: The discrete and continuous learners are trained in alternation, yet no stability or ablation analysis is presented. It would be valuable to understand whether one learner dominates or destabilizes the other during training.

Missing discussion on scalability and computational cost: Running two deep RL agents concurrently likely doubles the training cost and memory footprint compared to single-agent baselines (e.g., P-DQN). The paper should discuss these trade-offs explicitly.

Limited generality of the framework: While advertised as general-purpose, the system is tightly coupled with SB3, which constrains extensibility to other RL libraries or custom pipelines. The authors could better articulate why this framework should be preferred over dedicated PAMDP methods that might be more scalable.

Primarily an engineering contribution: The novelty lies mainly in software integration and experimental validation, with limited new algorithmic insight. The theoretical proof, while correct, is straightforward.

**Questions:**

How sensitive is performance to the discrete/continuous update ratio in the alternating training scheme?

Can the proposed approach scale to higher-dimensional continuous subspaces or actions with multiple parameters per discrete option?

---

### Official Review · Reviewer_b8aQ · 2025-10-30

**Soundness:** 3
**Presentation:** 3
**Contribution:** 2
**Rating:** 4
**Confidence:** 4

**Summary:**

The paper proposes a method to decompose parameterised action-space MDPs (PAMDPs) into equivalent standard MDPs so that off-the-shelf discrete and continuous RL agents can be combined to solve hybrid-action problems. The approach is formalized mathematically, with a proof of equivalence between a PAMDP and its decomposed MDP, and implemented as a modular framework compatible with Gymnasium and Stable Baselines3. The authors test various combinations of discrete and continuous algorithms (DQN, PPO, SAC, etc.) on the Platform and Goal environments, comparing them with baselines tailored to PAMDPs, namely Q-PAMDP and P-DQN.

**Strengths:**

- The work provides a theoretical equivalence proof showing that the decomposed MDP preserves the optimal policy of the original PAMDP.
- Offers a general-purpose framework to facilitate experimentation with hybrid-action RL problems.
- Provides an exhaustive description of the framework from an implementation perspective.
- Demonstrates that even without tuning, simple SB3 combinations can perform comparably to specialized algorithms in some settings. This favours the usability of the framework with off-the-shelf algorithms and avoids re-implementation.
- The paper is well written and clearly structured.

**Weaknesses:**

- **No repository link (for reproducibility):** The paper claims the code will be released, but no link is provided at the time of this review. Given ICLR’s strong reproducibility requirements, this omission is problematic. Moreover, since this work is strongly based on a framework contribution, public access to code is fundamental, and the lack of it severely limits the impact and verifiability of the work.
- **Lack of analysis of the alternating training scheme:** The authors train discrete and continuous agents in alternation, but no ablation or stability analysis is presented. How do the agents’ learning processes interact? Does one policy lag behind or destabilize the other? These interactions are critical to understanding convergence behaviour.
- **Lack of analysis on method scalability and complexity:** While the proposed solution is beneficial because it facilitates the use of already verified off-the-shelf algorithms in PAMDP problems, the authors do not evaluate the drawbacks of the proposed approach in terms of scalability and computational complexity. Training two deep RL agents simultaneously roughly doubles the memory footprint and computational cost compared to "single-agent" approaches such as P-DQN, as it requires storing twice the number of model parameters.
- **Limited “general-purpose” framework:** While the authors claim to provide a general-purpose framework offering a more flexible alternative to bespoke PAMDP algorithms (such as P-DQN and Q-PAMDP), the infrastructure is still tightly coupled with the SB3 library. Although SB3 is indeed one of the most used RL frameworks, this coupling limits the generalizability of the approach. Furthermore, there is no deep discussion of why one should prefer this framework over bespoke algorithms, which could be more scalable and less computationally demanding.
- **Limited and mostly engineering-focused contribution:** The theoretical proof is straightforward, and the main novelty lies in software design rather than algorithmic insight, making this work read as an engineering integration project — useful but not deeply novel.
- **No hyperparameter details:** Even though the authors state that no tuning was done for SB3 algorithms, hyperparameters are completely omitted also for P-DQN and Q-PAMDP. In particular, is not clear in the Platform experiment if the baselines have been tuned or not, since the Q-PAMDP's original paper shows different results. To facilitate reproducibility, reporting algorithms' parameters, learning rates, and training schedules is necessary and strongly recommended (ideally in an appendix).

**Minor Issues:**
- Please clarify the term “eNAC” (used in the Related Work section), which presumably refers to the episodic Natural Actor-Critic algorithm, and include the appropriate citation.
- Please use acronyms consistently. For example, "reinforcement learning" is sometimes written in full and sometimes abbreviated as "RL". Please ensure uniformity throughout the text.

**Questions:**

The following questions mainly derive from the previously highlighted weaknesses:
- How sensitive is the hybrid approach to the training schedule (i.e., the ratio of discrete/continuous updates)?
- Can the method scale to environments with multiple continuous parameters per action or high-dimensional continuous subspaces?
- How does the discrete-action learner actually learn if, as understood from the paper, it receives zero reward on its timesteps? What feedback drives its policy improvement?

---

### Official Review · Reviewer_USBh · 2025-10-31

**Soundness:** 2
**Presentation:** 2
**Contribution:** 2
**Rating:** 2
**Confidence:** 5

**Summary:**

Problem:

The paper addresses the problem of applying conventional RL algorithms to environments with parameterized actions.

Approach:

The authors propose to decompose the PAMDP problem into an MDP with double the horizon (alternating between selecting a discrete action and selecting parameters of the discrete action). This is done by augmenting the state space with (state, discrete action). The action set contains all discrete-continuous parameter combinations. Some actions become invalid in certain augmented states (the inconsistent discrete actions).


Overall assessment:

In my opinion, the paper seems more of a software engineering effort rather than a scientific contribution (writing gym-compatible code that works with standard RL implementations). My biggest problem is that the proposed approach may not fully exploit the mutual information between discrete and continuous choices that joint methods can leverage. The evaluation is also not sufficient: it includes only two domains and does not investigate any long-horizon domains. The impacts of doubling the horizon are not discussed. The paper points out that SOTA approaches perform well because of custom initialization, but what weight initialization strategy was used in the proposed approach remains unclear. The paper needs to address the open questions and improve on all of these fronts before it can be ready for publication.

**Strengths:**

- The modularity of the proposed approach stemming from using existing standard RL methods is useful.

- The paper raises important observations about the generalizability of the SOTA approaches without environment-specific tuning/customizations.

**Weaknesses:**

- The paper raises important questions about the generalizability of the SOTA approaches without environment-specific tuning/customizations. However, the paper does not really discuss how to address these problems.

- The policy decomposition forces discrete and continuous agents to operate sequentially, doubling time steps and possibly missing out on the benefits of optimizing both choices jointly. The proposed approach may not fully exploit the mutual information between discrete and continuous choices that joint methods can leverage. This is critical to addressing the inherent structure of parameterized action spaces.

- Initialization sensitivity: Some baselines (PDQN) show dramatic performance drops without domain-specific initializations. The paper’s hybrid framework might also suffer if initialization is not considered, though this is not deeply explored.

- Evaluation on just two domains is too few, especially when the proposed approach does not even perform well on one of them (the Goal domain). Moreover, a more thorough investigation is required by evaluating domains with longer horizons. It is unclear how the approach would perform in diverse or real-world applications outside these benchmarks, especially where decision-making is required for longer horizons.

**Questions:**

- What are the limitations of doubling the horizon for decision-making? How does this affect the complexity?

- In tasks where the association between discrete actions and their parameters carries mutual information, joint approaches can exploit these connections better. What are the author's thoughts on this?

- What weight initialization strategy was used in the experiments of the proposed approach?

- The paper does not discuss how to address the problems of limited generalizability. What is in the proposed approach that addresses this?

- Minor: Why are two different names mentioned for the same domain in the abstract (goal and robot soccer)? This could cause confusion.

Refer to the weaknesses section for more questions.

**Details Of Ethics Concerns:**

The name of the University from which the GPU resources are used is specified (line 338). Not sure if this is against the policies.

---

### Meta-Review · Area_Chair_fbxF · 2026-01-09

**Summary:**

- The primary concerns were around the lack of comparison with baselines (e.g., DLPA, HyAR), limited evaluation domains and lack of details.
- The authors also did not write a rebuttal

**Reviewer Concerns:**

See above.

**Reviewer Scores:**

General concerns and consensus that the paper lacked scientific novelty and had insufficient evaluations.

---

### Decision · Program_Chairs · 2026-01-26

Reject